# An Effective and Efficient Adaptive Probability Data Dissemination Protocol in VANET

**John Sospeter** [1] , **Di Wu** [1,*], **Saajid Hussain** [1] **and Tesfanesh Tesfa** [1]

School of Computer Science and Engineering, Dalian University of Technology, Dalian 116024, China;
jsospeter0@gmail.com (J.S.); sunny_sau@hotmail.com (S.H.); tesfanesh21@yahoo.com (T.T.)
* Correspondence: wudi@dlut.edu.cn; Tel.: +86-138-9861-7601

**Abstract:** Mobile network topology changes dynamically over time because of the high velocity of vehicles. Therefore, the concept of the data dissemination scheme in a VANET environment has become an issue of debate for many research scientists. The main purpose of VANET is to ensure passenger safety application by considering the critical emergency message. The design of the message dissemination protocol should take into consideration effective data dissemination to provide a high packet data ratio and low end-to-end delay by using network resources at a minimal level. In this paper, an effective and efficient adaptive probability data dissemination protocol (EEAPD) is proposed. EEAPD comprises a delay scheme and probabilistic approach. The redundancy ratio ($r$) metric is used to explain the correlation between road segments and vehicles' density in rebroadcast probability decisions. The uniqueness of the EEAPD protocol comes from taking into account the number of road segments to decide which nodes are suitable for rebroadcasting the emergency message. The last road segment is considered in the transmission range because of the probability of it having small vehicle density. From simulation results, the proposed protocol provides a better high-packet delivery ratio and low-packet drop ratio by providing better use of the network resource within low end-to-end delay. This protocol is designed for only V2V communication by considering a beaconless strategy. the simulations in this study were conducted using Ns-3.26 and traffic simulator called "SUMO".

**Keywords:** data dissemination; road segments; vehicles' density; safety message application; broadcast storm problem

## 1. Introduction

A vehicular ad hoc network (VANET) consist of moving vehicles connected via wireless technology e.g., Wireless Access in Vehicular Environment (WAVE) for the aim of exchanging information. The vehicles are considered as "nodes" within a network [1]. VANET is the subclass of the mobile ad hoc network (MANET), and the difference between them is that MANET is mobile-based whereas VANET is not. Since the movement of vehicles in VANETs is of varied speed, the design of the data dissemination protocol is considered an important aspect in this field of researches. The primary goal of VANET is to increase the safety of the passengers.

Communication in VANET involves vehicles and other infrastructure along the roads designated as the "on board units" (OBUs) and "roadside unit" (RSU), respectively. These units are hardware devices that ease the exchange of data within the vehicular environment. Communication among the hardware's can be classified into three categories as depicted in Figure 1. Vehicle-to-vehicle (V2V), this communication involves vehicles directly communicating with each other without depending on other infrastructure. The purpose of V2V is to support the data dissemination application relying on the application of the IEEE 802.11p standard [2]. The vehicle-to-infrastructure (V2I) includes

vehicles communicating with roadside infrastructures for the purpose of collecting and gathering data. The hybrid architecture is the combination of both V2V and V2I whereby vehicles and roadside infrastructures communicate with each other in different ways depending on the distance parameters.

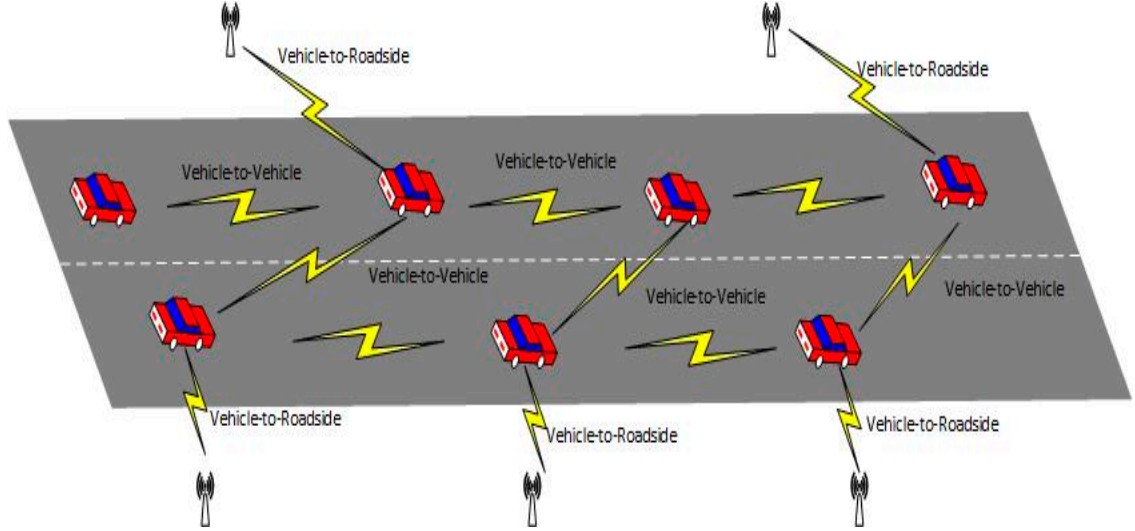

**Figure 1.** Vehicular ad hoc network (VANET) communication architecture.

The main challenge is a high number of vehicles moving with high speed, and therefore it takes a short period of time for reliable exchange of data within a network (connectivity problem). Another challenge faced in this technology is the restriction of road structure and traffic issues in the vehicular environment. Such activities often bring new challenges in VANET. Hence the study of data dissemination in VANET should be taken into consideration.

Many researchers have specified two types of VANET application such as the safety and no-safety applications. However, the most important is safety application since it plays a significant role in reducing the number of accidents as the major goal of design VANET [3]. This application gives the driver an early warning (alert message) of an accident that has occurred. This helps to prevent accidents by giving the drivers extra time to react. Nevertheless, the VANET application can use the available network resources such as bandwidth to supply other services (non-safety application). The safety application includes valuable information, advertisements, and entertainment. The role of this application is to comfort the passengers and provide other applications such as parking availability services, and internet connectivity [4]. The entertainment application like sharing movies, music, chatting with each other during their long journey is not linked to the safety application.

The data dissemination process in V2V depends on the application of IEEE 802.11p standard [2]. This vehicular environment can tolerate limited network bandwidth, thus the bandwidth of 10 MHz is divided into six service channels (SCH) as well as one control channel (CCH) CCH is designed to provide safety applications only in a vehicular environment. The design of the data dissemination protocol should fit the different requirements of various VANET applications such as improve reliability, decrease dissemination delay time and increase the packet delivery ration [5–8]. Many broadcast techniques especially traditional ones such as blind flooding originate from the "broadcast storm problem" [9]. This happens when every node within a network area must rebroadcast a packet immediately after receiving the message to ensure the message is delivered to distant vehicles in the network. By doing so, a large amount of duplicated messages are disseminated. This leads to a huge amount of bandwidth utilization and, when the network density is increasing especially in the urban environment, it leads to high channel contention and high collisions.

In this paper, a data dissemination protocol was proposed, the effective and efficient adaptive probability data dissemination protocol (EEAPD).

The main contributions of this work are fourfold as follows:

1.  EEAPD delivers an effective and efficient data dissemination protocol. The proposed protocol is more useful because it considers the critical parameters such as the number of road segments (slots), minimal waiting time, number of vehicles (vehicles' density), and message direction in rebroadcasting probability decisions. EEAPD ensures effectiveness and efficiency by not only evaluating the low end-to-end delay but also high packet delivery. Additionally, it offers effective delivery of a message for the nodes' receiver to understand, interpret and make use of it as intended. Nevertheless, the EEAPD protocol reduces bandwidth consumption.
2.  EEAPD provides the ability to adapt to the protocol behavior efficiently for safety applications' requirements. This protocol is an adaptive probability protocol that offers efficiency and effectivenesses for safety applications. This application is intended to ensure the passengers' protection from danger, risk or injury on roads.
3.  EEAPD offers the ability to adopt an environment with no beacon exchange. The protocol depends only on surrounding vehicles' density within road segments.
4.  The simulation of EEAPD considered the number of road segments and vehicles' density. EEAPD was compared with beaconless protocol and beacon-based protocol. Results show that EEAPD enhanced the performance of data dissemination in a realistic environment.

The rest of the paper is systematically arranged as follows. Section 2 provides related works on the data dissemination protocol. Section 3 provides more explanation about the EEAPD protocol. Section 4 describes the simulation settings, as well as discussion of performance assessment were provided. Section 5 provides concluding remarks and future works.

## 2. Related Works

Generally, there are many techniques suggested by different authors to explain the concept of data dissemination in VANET relying on transmission strategies such as unicast, multicast and broadcast [10–15]. These different techniques have been explained in different studies [16–18]. Some studies are specifically designed for V2I communication architecture and others for V2V architecture. In data dissemination, V2V broadcasting data is considered the most important technique. However, this technique suffers from the "broadcast storm problem" where there is a huge number of vehicles (dense area) as a result of a large amount of information exchanged in the network area. Many researchers have suggested different techniques to overcome the broadcast storm problem. Some of these techniques considered as delay-based include slotted 1-persistence, weight p-persistence and slotted p-persistence.

Basically, according to the slotted 1-persistence protocol (S1PD) [19], the transmission range is divided into various areas designated "slots". Whereby each slot is given a different timer variable designated "timer slot". The slot which will be farthest away from the source node will be given the shortest time slot. The vehicles belonging to the furthest slot are given the shortest waiting time to rebroadcast the message. The node starts decrementing its time instantly after receiving the new message. Then after the expiration of time, the vehicle rebroadcasts the message provided that there is no redundancy in the received messages during waiting time. Besides, vehicles allocated to other slots are supposed to cancel the transmission as soon as they receive a redundant message. As a result, the "redundant" broadcast problem decreases. Eventually, S1PD reduces uncontrolled redundant rebroadcast by providing high-packet delivery as well as low end-to-end delay.

Apparently, the S1PD protocol may undergo synchronization problem as explained in [20]. This happens when large numbers of vehicles allocated on the same segment with the same timer begin their broadcast concurrently leading to the collision problem. Therefore, Schwartz et al. [21] propose another protocol named "Optimal 1-Persistence Dissemination" technique [O1PD]. O1PD is the successor of S1PD which tries to solve synchronization problem in S1PD protocol. This happens by increasing the amount of delay time allocated to the network layer whereby each time segment

allocated to each node in O1PD should correspond to its distance to the source node and operate towards its message direction.

"Surrounding vehicles' density" is a technique that solves synchronization problem during the transmission process [22–26]. Schwartz et al. [22,23] proposed a protocol called "distributed optimized time (DOT)". This protocol is an improvement of O1PD based on surrounding vehicle density in the transmission range. DOT tries to distribute a number of nodes allocated to the same slot by minimizing redundancy problem. Moreover, a probabilistically based protocol was proposed whereby the probability of broadcasting "P" depends on the surrounding vehicle density [24]. In 2010, Tseng et al. developed a scheme to customize the number of road segment depending on the surrounding vehicles density [25].

A hierarchical routing scheme designed for a "multihop" network known as the cluster-head gateway switch routing (CGSR) protocol was proposed by Chiang [27]. in this protocol, nodes are grouped into a clustered multihop. The cluster-head node controls and manages other nodes belonging to the specific cluster. By considering cluster-head nodes, CGSR enhanced routing efficiency since these nodes have the power to manage the transmission medium and forward packet data to gateway nodes. When either gateway or cluster-head node turns to the normal node, it is supposed to forward all packets to its cluster-head, and at this point the routing become normal. The disadvantage of this protocol is how to maintain its clusters since each node needs to update its table from time to time when receiving updates. Although the CGSR transmission delay is reduced to 32,238, to run this scheme incurs significant overheads.

Lin et al. [28], introduce a different degree of synchronization, where the cluster-heads control data packet routing and forwarding in order to handle channel scheduling e.g., The Code Division Multiple Access/Time Division Multiple Access (CDMA/TDMA) techniques. The problem with these techniques comes when assigning a code to a different cluster and cluster-head focus on the traffic of a cluster.

Edison et al. [29], propose an approach in which a software agent is used to gather data from the specified area called "target region (TRs)" and to disseminate data in MANET and VANET. Software agents use context geographical information about the node (position information e.g., current location and future location, direction information, and route to reach destination) to make a decision on migration among the mobile nodes in the network. In this work, the authors address an important feature on how these software agents may use a different technique and decision parameter to conduct their movement in the network. The work proposed three categories of approaches depending on the different levels of intelligence on the conduct of the agents' movement decision. First, the destination-based approach, whereby a software agent determines whether the hosted node is inside or outside the target region and whether the future destination of a node is inside or outside the target region. Second, for this level of intelligence, the software agent has both information from the course, hosting node position and the destination node using a global positioning system (GPS). Lastly, the direct path approach, in which the agent is considered to have the complete route of the hosted node and destination node that keeps the agent inside the target region. However, in this mechanism, the protocol does not indicate the reality of data dissemination in terms of message delivery and the end-to-end delay of software agent migrating from one node to another.

However, in [30] authors propose a simple and efficient adaptive dissemination (SEAD) protocol in the vehicular environment. SEAD considers message direction and vehicles' density to decide the probability of broadcasting. This protocol also considers no beacon exchange in neighbors' information such as speed, location, angle etc. Upon receiving a packet, the nodes allocated to the farthest road segment are assigned minimum $W_t$, then rebroadcast after the expiration of $W_t$ ($0\delta$). As the time number of nodes allocated to the same slot increases, the probability of re-broadcast is calculated on advanced terms, but this protocol does not explain in detail the impact of road segments on rebroadcasting probability in term of high vehicle density and low vehicle density. Particularly,

there is a fixed parameter ($\alpha$) which helps to manipulate the packet delivery ratio depending on the application's requirements.

In delay-based [31] systems, authors examine two techniques of this protocol, namely slotted and continuous techniques in a vehicular environment. This strategy every receiving node depend on a certain waiting timer denoted $W_t$ before decide to rebroadcast the message. Authors examine these techniques to evaluate the performance of a "redundancy-based protocol" (RBP) [32], which is the combination of probability-based and delay-based. For every calculation of waiting timer, rely on deployed technique. The authors consider a slotted technique in a dense network when the nodes in the single segment increased and try to rebroadcast the message simultaneously, which may result in message collision. This technique has the following features; first, when two nodes are close to each other but belonging to different slots they have enough time to exchange the received message. Second, depending on the vehicular environment, in order to handle the lost messages, the technique takes into account a certain amount of message redundancy. In the continuous technique, taking into account the distance from the source node to the receiver node and farthest vehicle having the shortest time has the chance to rebroadcast the message in the message direction. This technique has the highest probability to reduce the number of forwarder nodes in the network environment, thereby helping to resolve the collision problem in the high dense network. In the end, authors propose three versions of RBP namely; RBP with a linear continuous dissemination protocol (RBP-LCD), RBP with a non-linear continuous dissemination protocol (RBP-NLCD), and RBP with the slotted protocol (RBP-S1PD). However, the continuous technique does not fit on a safety application

Oliveira et al. [33] proposed the adaptive data dissemination protocol (AddP). This is kind of Multi-hop broadcasting protocol that considers the distance from neighboring vehicles and local density on candidate selection procedure. To reduce channel contention and number of redundant message the AddP use candidate selection procedure. When one vehicle fails to rebroadcast the message, the neighbor's ones should do it. The proposed protocol is based on beacon exchange in order to know the neighboring information e.g., position, angle, local density and speed. The proposed protocol uses warning message aggregation by providing network coding strategy to minimize the amount of warning message rebroadcasted. On data collection sent by nodes, the road side unit (RSU) acts as a point of collection. RSU uses the dissemination messages monitoring procedure to investigate the last message received by a node, short-time connectivity and addresses the hidden node problem in the current network. Moreover, AddP continues to suffer from high message delay since it does not show the direction of message propagation. Authors address this problem as a future work.

Now, the study of the Internet of Things (IoT) is growing very rapidly and many researchers evaluate how the objects (stationary and mobile) can exchange and share data in the area of interest. VANET is one of the technologies can relay in IoT whereby vehicles and road infrastructure can communicate with each other by different means wireless technology e.g., WAVE. Authors in [34,35], explain the idea of how different entities such as interest, physical and energy-aware in the clustering and allocating resource procedure in a machine-to-machine (M2M) communication network. This helps the adaptability of communicating different devices such as V2V in VAVET based on IoT. However, the authors in [34,35] did not consider the mobility of the nodes.

The proposed protocol differs from previous protocols as explained in this paper whereby a proposed protocol explains how a number of vehicles should be allocated to a single road segment by increasing many road segments according to vehicle density; with a small number of vehicles almost one vehicle to be assigned to a single road segment may result in overcoming the synchronization problem and avoid a collision. Nevertheless, in this paper, the redundancy ratio parameter was clarified which helps to control the rebroadcasting probability depending only on neighboring vehicles' density by considering no beacon exchange.

Table 1 shows a comparison of selected protocols by examining the following features: (a) application type; (b) forwarder parameter, used to select the forwarder node; (c) simulated scenario, part of the scenario simulated; (d) beacon-based; (e) communication mode. The protocols involve

(highlighted) to evaluate the comparative performance with proposed protocol are beacon-based such as S1PD and AddP as well as beaconless (SEAD and delay-based).

**Table 1.** Related works comparison.

| Selected Work | Application Type | Forwarder Parameter | Simulated Scenarios | Beacon-Based | Communication Mode |
|---|---|---|---|---|---|
| **S1PD** [19] | Safety | Distance | Highway | Yes | V2V |
| O1PD [21] | Safety | Distance | Highway | Yes | V2V |
| DOT [22] | Safety | Distance and Density | Highway | Yes | V2V |
| **SEAD** [30] | Safety and Non-safety | Density, Distance and Msg direction | Urban and Highway | No | V2V and V2I |
| **Delay-Based** [31] | Safety | Density and Distance | Urban | No | V2V and V2I |
| **AddP** [33] | Safety | Density and Distance | Urban and Highway | Yes | V2V and V2I |
| EEAPD | Safety | Segments, Distance, Density and Msg direction | Urban | No | V2V |

Msg: Message.

## 3. Effective and Efficient Adaptive Probability Data Dissemination Protocol (EEAPD)

In this paper, an accurate hybrid based dissemination protocol was suggested, EEAPD. This protocol is a combination of probabilistic and delay mechanism based on the number of road segments and distance from the source node. The purpose is to produce a desired or intended result (effectiveness) and achieve maximum productivity with a minimum amount of bandwidth consumption (efficiency) in high network density. The proposed method minimizes the broadcast storm problem and at the same time provides low end-to-end delay and high data packet delivery in a highway environment. The most important thing is that the distance variable was used for calculating the waiting time period. The EEAPD protocol considered the vehicles' density, messages' direction, and a number of road segments (slots). Additionally, it offers the least waiting time for a vehicle to determine which node is suitable for a re-broadcast message by considering its probability. In the proposed protocol there are two key features: firstly, EEAPD is beaconless, that is, no periodic safety message exchange is considered even if the vehicles' density is examined. Secondly, the proposed protocol is only suitable for a safety application.

To gain more understanding of the proposed protocol, the basic premises and assumption are explained in detail. Then, the mutual relationship between the vehicles' density and "redundancy ratio" was given. Finally, a more detailed explanation about EEAPD protocol and essential steps is given in Section 3.

### 3.1. Requirements and Premises

In this paper, only vehicle-to-vehicle communication as depicted in Figure 1 was contemplated by assuming each node is provided with onboard unit device having IEEE 802.11p standard. Likewise, every node can deduce its current position by considering GPS. This utilization is accountable for generating an emergency message that needs to spread within the network area. In this paper message and data packet will be used interchangeably.

### 3.2. Redundancy Ratio Metric

The redundancy ratio "$r$" is a reliable metric used to measure the proportion between the number of delivered messages per total new messages. It is calculated by:

$$r = \frac{TotalReceivedDataPacket(original + redundant)}{TotalNewDataPacket(original)} \tag{1}$$

From the proposed design, during the entire message transmission process, each node repeatedly updates its redundancy ratio "*r*" within the specified time $\Delta t$. The redundancy ratio can be reset when there is no data packet being received. From Figure 2 the simulation experiment was performed by a fixed number of the source node to show the correlation between a number of vehicles and its corresponding redundancy ratio (more explanation is provided in Section 4). Figure 2 shows that redundancy ratio is directly proportional to vehicles' density. This indicates that when the vehicles' density becomes higher, the number of redundant packets also increases. Therefore, for surveillance, it is clearly logical commence that in EEAPD the rebroadcast probability depends on the redundancy ratio, such that the more redundancy ratio becomes high the less the probability of rebroadcast, and vice versa. This helps to reduce the broadcast storm problem when the number of nodes decreases.

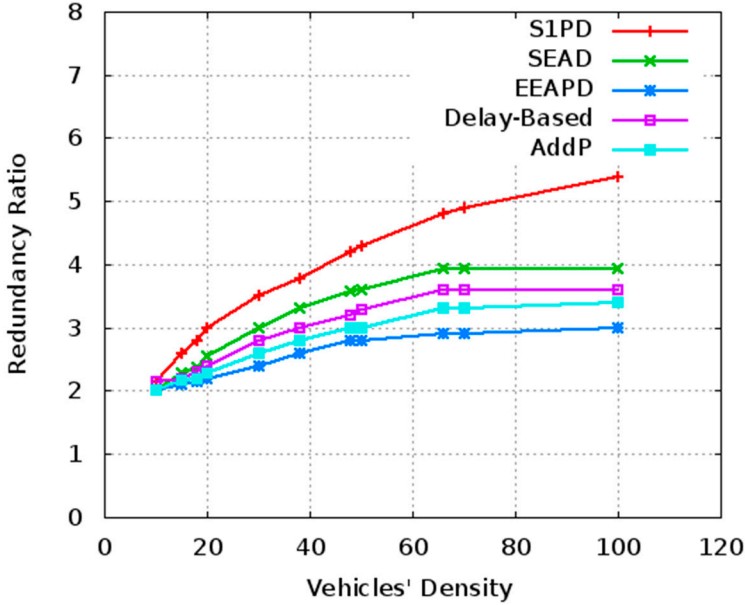

**Figure 2.** Redundancy ratio vs. number of vehicles.

### 3.3. EEAPD Protocol Description

The primary idea of EEAPD is shown by using two flow chart diagram presented in Figures 3 and 4. From these figures, the explanation is divided into three phases and is described as follows. Each event-driven message (emergency message) explained in the EEAPD protocol (Figure 5) is distinguished by a special identification whereby this identification includes the source vehicle's identification and message identification, assuming a message's header consist of both broadcasting vehicle identification and its position by GPS. Every node posse's data buffer which is special for kept the received original messages except the redundancy ones. EEAPD checks if the packet received by the vehicle is recognizable or not by considering the packets identification kept in the data buffer.

EEAPD has three phases: firstly, if the message is not known (new message) only received message coming in front of vehicle message direction is considered to be spread in the network area, as stated by EEAPD on the other hand, the message (new data packet) is kept in buffer by a name "*schedule for rebroadcast* = *True*" then regard this message as having a special rebroadcasting by considering the waiting timer "$W(t)$" then immediately trigger. Nevertheless, packet data coming from other direction except in front of it show as the reachability from the message direction, this message given by a name "*schedule for rebroadcast* = *False*".

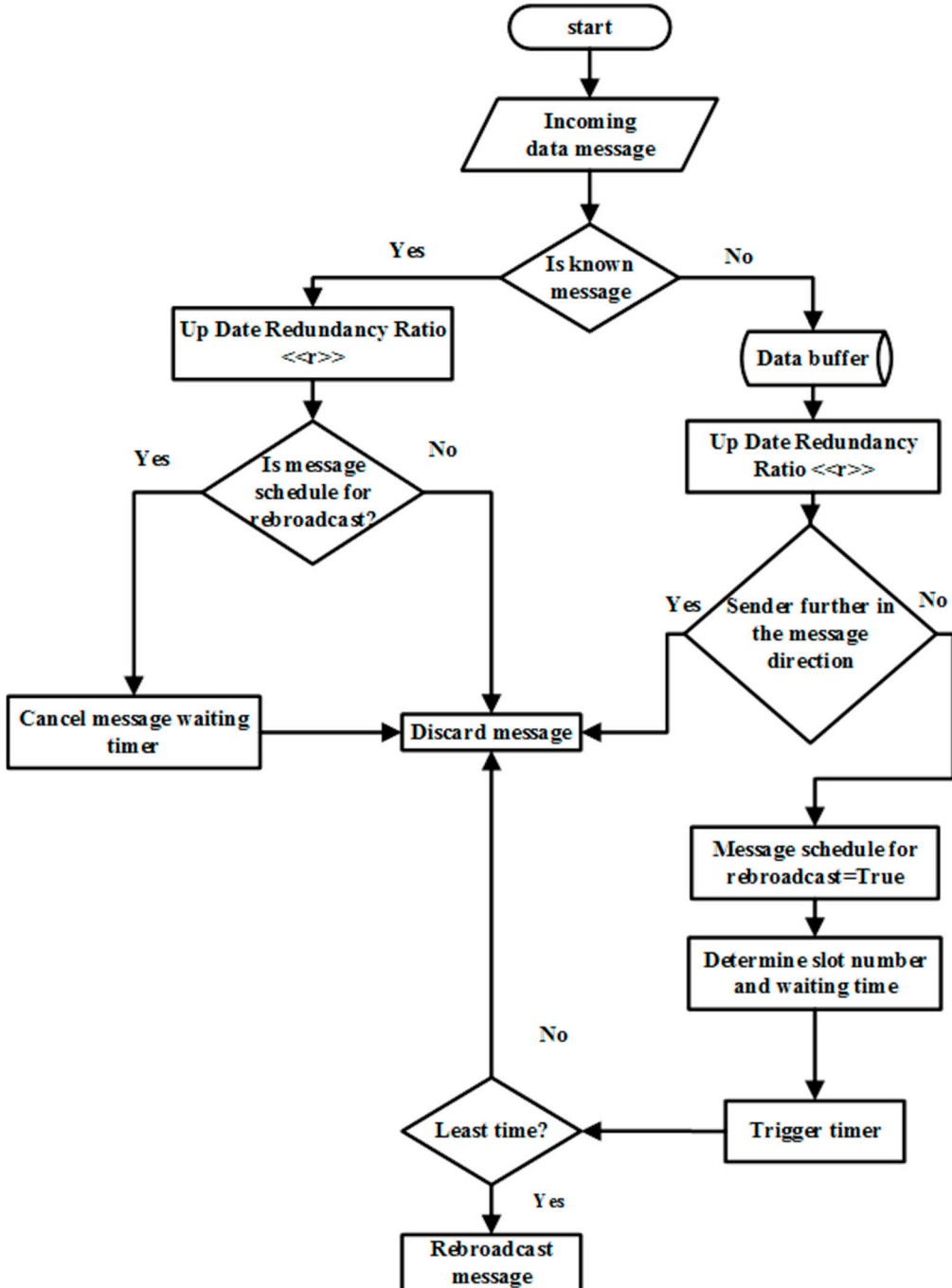

**Figure 3.** Receiving procedure of effective and efficient adaptive probability data dissemination protocol (EEAPD).

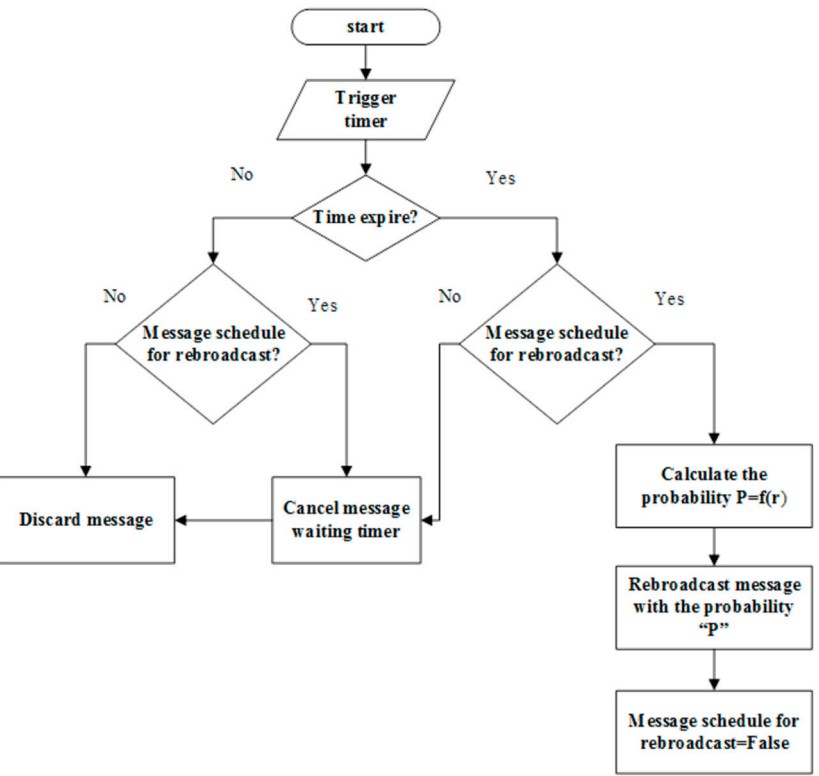

**Figure 4.** Rebroadcasting procedure of EEAPD.

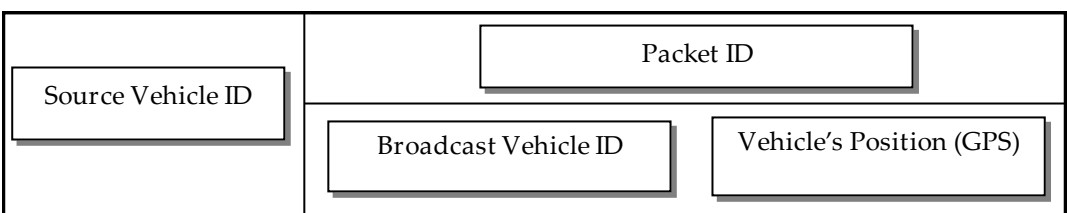

**Figure 5.** Event-driven packet format.

Secondly, the message timer should continue to decrease and when it expires with no duplicated received, the trigger timer will determine if it least time among road segments, if this is the case then the message is broadcasted with probability "$P = f(r)$". Otherwise, cancel both messages waiting for the timer and rebroadcast decision. Thirdly, when the message is known (not new) its waiting timer period "$W(t)$" should instantly be canceled and discarded if it is scheduled for rebroadcasting, else it is discarded. In both these phases, the redundancy ratio "$r$" should be updated at the receiving event.

The idea implemented in EEAPD is motivated by different protocol schemes such as S1PD and SEAD. By considering the number of segments (slots) that are fixed, waiting time can be computed as shown in Equation (2).

$$W_t = \left[ N_{st} \cdot \left( 1 - \frac{min\left( D_{ij},\ R \right)}{R} \right) \right] \cdot \delta + \mu delay \tag{2}$$

such that; $N_{st}$ is a number of road segments (dynamically selected), $D_{ij}$ is the respective distance from source node "$i$" to receiver node "$j$", $R$ is mean transmission range and $\delta$ is estimated 1 hop delay consists of medium access delay and propagation delay.

For more explanation about this strategy, an example of the system model mechanism is depicted in Figure 6. Assume the source vehicle (S) transmit the emergency accident message about 300 m in

transmission range. With respect from the source node, all nodes (vehicles) within network area are allocated with different slots.

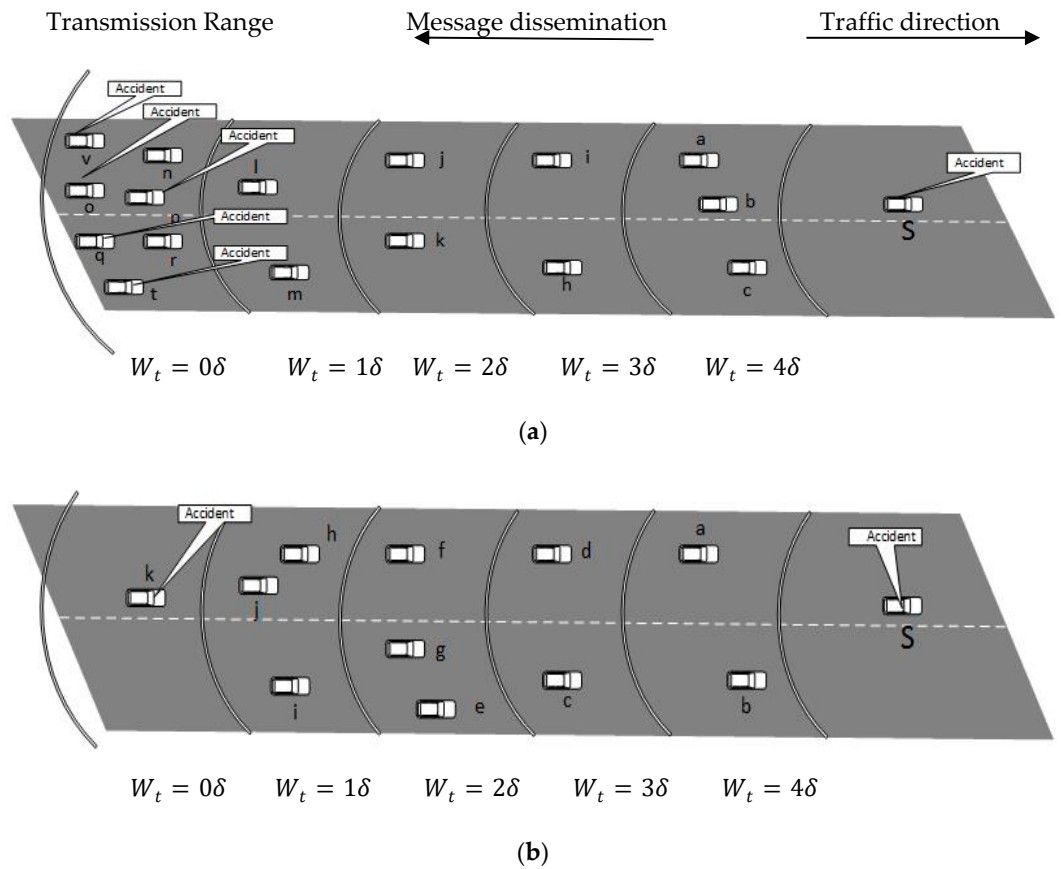

**Figure 6.** The system model for EEAPD, (**a**), Many vehicles at the last road segment; (**b**), Few vehicles at the last road segment.

Impact of Roads Segment on Rebroadcast

In this case, (refer to Figure 6b). when the number of vehicles increases, the redundancy ratio (*r*) also increases hence, the probability of rebroadcasting become small. From this concept, it is clearly seen that the decreasing of vehicles' density by giving a greater number of road segments in the transmission range leads to small redundancy ratio and finally the probability of rebroadcasting for suitable nodes increasing. Assuming that the Dalian University of Technology (DUT) map (refer to Figure 7) there are numbers of road segments (slots) within transmission range, each road segments (slots) i.e., one slot having almost 100 m apart to each other (refer to Figure 6). This results in low vehicle density in each road segment (Refer to Figure 8).

As specified in Equation (2) holding time is inversely proportional to the distance from source nodes to receivers' nodes. Therefore in Figure 6a. (vehicles o, p, q, r, t, v, and n) belonging to the farthest road segment are assigned to the minimum $W_t = 0\delta$. (Vehicles l and m) belong to the same slot must wait for $1\delta$ where (vehicles j and k) have to wait for for $2\delta$, (vehicles h and i) have to wait for $3\delta$, and (vehicles a, b and c) have to wait for $4\delta$ prior to transmitting. This value ($\delta$) is an important time factor that makes it responsible for a node to wait from when it receives the data packet to before it decides to retransmit to other nodes. According to this mechanism, only members' vehicles allocated to the last slot on transmission range will be chosen as unique forwarder vehicles. Therefore, assigning a different timer to different slots contains nodes will help to minimize the broadcast problem. In spite of that, in a dense area where there is a large number of vehicles allocated to the same slots with the same given timer the problem can happen. As found in Figure 6a. at the same time (vehicles o, p, q, t,

and v) immediately will rebroadcast the received data packets. In this case, no collision can happen. But the problem will occur if a number of nodes allocated in the same slots becomes high.

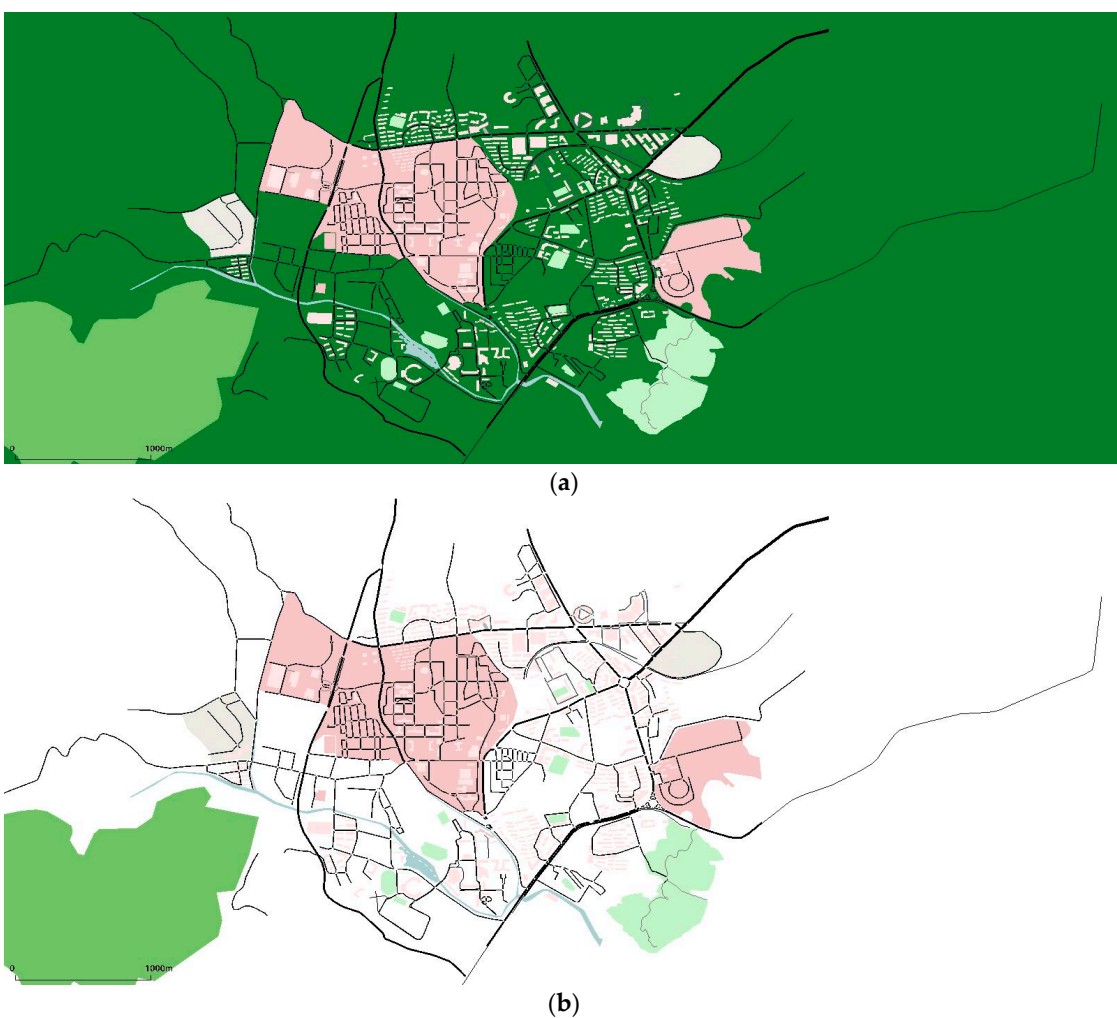

(**a**)

(**b**)

**Figure 7.** The Dalian University of Technology area, (**a**) real world map; (**b**) standard map.

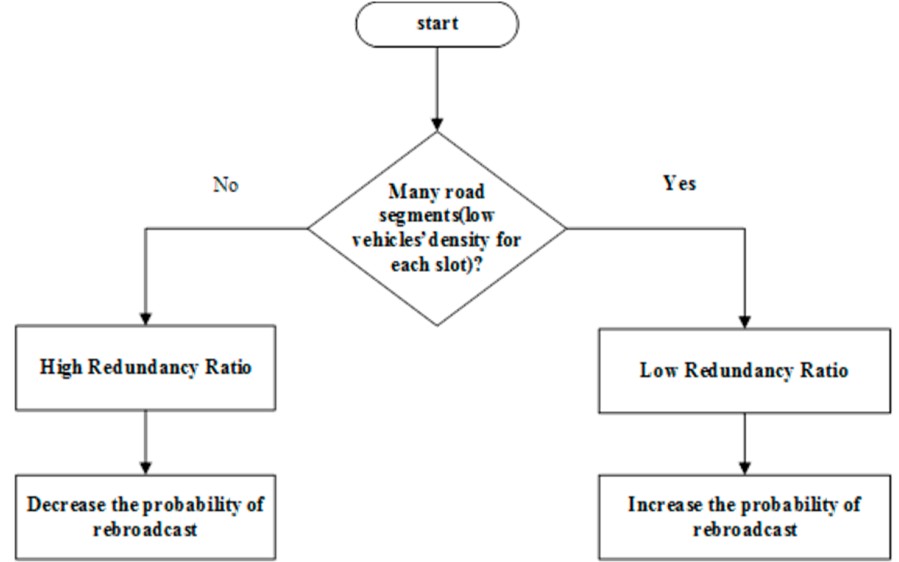

**Figure 8.** The relationship between vehicle density, redundancy ratio and the probability of rebroadcast.

To handle this complicated situation, an efficient and effective method of computing the rebroadcasting probability, namely "*P*", was designed. EEAPD has the ability to work without the need for beacon exchange (neighbor vehicles information) but it only considers enclose vehicle density. The probability, given by Equation (3), is the association of present redundancy ratio values "$r_i$" computed at dispatch time, and preceding one, named "$r_{prec}$" by including the preceding computed probability "$P_{prec}$" for the final sent data packet. The motivation behind this relationship is to control the redundancy ratio.

$$P_1 = \frac{2}{r} \cdot P_{prec} = \frac{2}{r_0} \cdot \frac{2}{r_{prec}} = \frac{2}{r_0} \cdot \frac{2}{r_1} \cdots \frac{2}{r_{i-1}} \cdot \frac{2}{r_i} = \frac{(2)^{r+1}}{\Pi_{k=0}^{i} r_k} \tag{3}$$

Thanks to EEAPD, as shown in Figure 6b, the vehicle (*k*) found in the last road segment was selected to rebroadcast the received message. Due to that, there are many road segments in the transmission range and the last slot found that there is only one node needed to rebroadcast the message. It is shown that the lower the number of vehicles the smaller the number of duplicate messages. Therefore, these increases rebroadcast probability opportunity as shown in Figure 6. In Figure 8, the probability of a vehicle to forward a data packet is opposite to both the redundancy ratio and the number of vehicles. Hence, the road segment which has a large number of vehicles will diminish the vehicle's satisfaction with being a forwarding node.

## 4. Performance Assessment

### 4.1. Simulation Platform

Based on the simulation conducted in the vehicular environment, the performance in efficiency and effectiveness of EEAPD was analyzed. The experiment was conducted using a Ns 3.26 [36] simulator. By considering the simulated map, the realistic mobility pattern was generated via micro traffic simulation named "simulator of urban mobility" (SUMO) [37], where the map of Dalian University of Technology (DUT) in China presented in Figure 8 constitutes many sets of road segments, for each slot within communication range is multilane road. The number of the lane may vary from 2 to 4 lanes. Simulation setting parameters are outlined in Table 2.

**Table 2.** Simulation framework parameters.

|  | Parameters | Specification |
|---|---|---|
| Realistic scenario | Network simulator | Ns-3.26 |
|  | Mobility generator | SUMO |
|  | Simulation duration | 200 s |
|  | Deployed Map | Dalian University |
|  | Road length | 5000 m |
|  | Number of source nodes | 5 |
|  | No. of Vehicles (Veh/km) | 10 to 100 |
|  | Propagation model | Nakagami |
| Application | Max speed | 20 m/s |
|  | Packet frequency | 1 Hz |
|  | Packet size | 500 bytes |
|  | No of run | 20 |
| Physical Layer IEEE 80.11p | Radio band | 5.9 GHz |
|  | Bandwidth | 10 MHz |
|  | Transmission range | ~700 m |
| Delay-based Technique | $\delta$ | 4 ms |
|  | $N_{st}$ | From 150 to 450 |
| Media Access Control Layer | bit rate | 6 Mbps |

Four dissemination protocols are chosen for the performance comparison, namely:

- AddP: this is the protocol that considers the beacon exchange in rebroadcasting message according to its distance from the source node and network density [33].
- Delay-based: this protocol based into two strategies namely continuous strategy and slotted strategy, the originality of this protocol is come from the redundancy-based protocol (RBP) [31].
- SEAD: upon receiving a packet, the members' nodes allocated to the last slot are assigned minimum waiting period of time "$W_t$" then rebroadcast it after the expiration of $W_t$, if number of nodes owned by same road segment is higher, the probability of re-broadcast is calculated on advanced. This protocol depends on distance, message direction and vehicles' density no beacon exchange [30].
- S1PD: during receiving a message, the node examines the originality of the message, after waiting time $W_t$ become invalid (expire) then it rebroadcast according to [19].

To assess the performance of the EEAPD protocol, the following criteria were examined:

- Forwarding Ratio: the ratio of nodes participates in the rebroadcast of a source message.
- Packet Delivery Ratio: the mean amount of original message accomplished received by a node with respect to the total amount of produced messages.
- End-to-End Delay: the period of time taken for a message to be broadcasted from source node to the last receiver node (destination) in the network.
- Packet Drop Ratio: the mean faulty of accepted delivery of packets of a node with resembling the total amount of accepted delivery of the packet.
- Link Load (bit/s): the mean amount of broadcast congestion (b/s) accepted by a node per period of time.
- Relaying vehicle (%): number of forwarder vehicle per total number of vehicles participated during simulation.

### 4.2. Adaptation of Proposed Protocol in the Realistic Environment

With the help of probability formula for broadcasting data (refer to Equation (3)), each node can decide when to rebroadcast the accident message depend on the current state of the network. The performance of the EEAPD protocol to the networking environment is adopted on the safety application requirement by considering two parameters namely "road segments" and "vehicle density". For better proof of the correctness of EEAP adaptability in a realistic network environment, we demonstrate the proposed protocol under both parameters in which the higher data reachability obtained around 98% and 97% on both road segments and vehicle density respectively while saving network capacity and limited bandwidth consumption.

### 4.3. Impact of Road Segments in the Reliability of Data Dissemination

In this part, the impact of road segments (slots) in relation to the packet delivery ratio, redundancy ratio, end-to-end delay and relaying vehicles achieved was examined when deployment of EEAPD protocol to a networking environment. The adaptability of EEAPD protocol safety application requirement according to how a number of road segments parameter can be examined through Figures 9–12.

Figure 9 demonstrates the relationship between redundancy ratio according to the road segments. It shows clearly that the redundancy ratio decreases better with the increase of road segments in EEAPD from road segments is 300, the redundancy ratio decreases to 20% and improves to 5% at road segments equal to 450, which is close approximately to a 0% redundancy ratio (no more redundancy ratio in data dissemination). However, using SEAD, S1PD, Delay-Based and AddP protocols, the redundancy ratio decreases and reaches a constant state when the road segments increase from 360 to 450 (refer to Table 3).

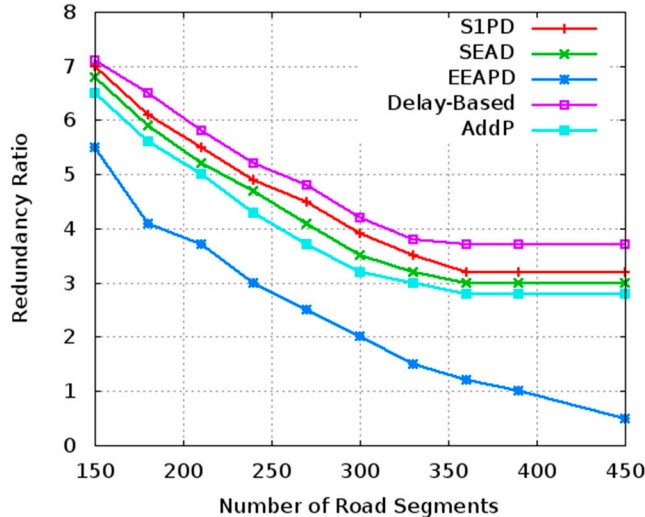

**Figure 9.** Number of road segments vs. redundancy ratio.

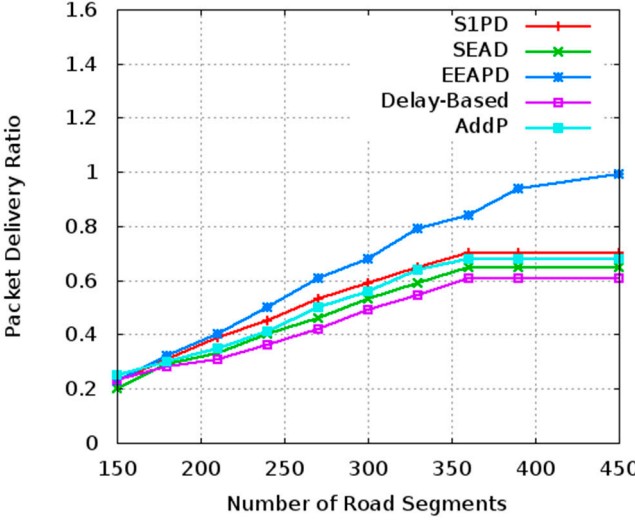

**Figure 10.** Number of road segments vs. packet delivery ratio.

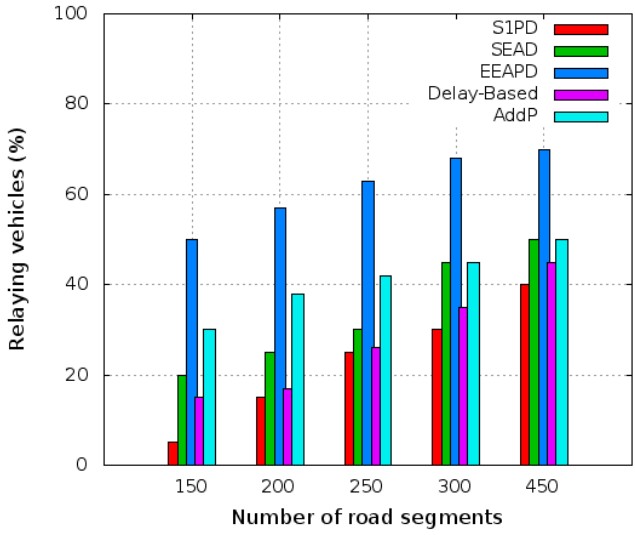

**Figure 11.** Number of road segments vs. relaying vehicles.

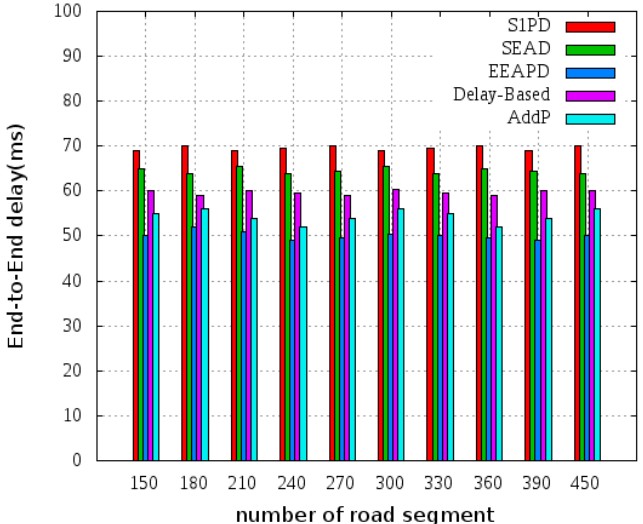

**Figure 12.** Number of road segments vs. end-to-end delay.

**Table 3.** Protocols comparison based on number of road segments.

| *PROTOCOL* | **Rr (%)** | **PDr (%)** | **Rv (%)** | **E2E (ms) at 450 Segments** |
|---|---|---|---|---|
| S1PD | 32 | 70 | 40 | 70 |
| SEAD | 30 | 65 | 50 | 64 |
| Delayed-Based | 37 | 65 | 50 | 64 |
| AddP | 28 | 68 | 50 | 56 |
| **EEAPD** | **5** | **98** | **70** | **50** |

This means that a larger number of segments generate a smaller number of vehicles in single slot consider that each slot is far apart to each other enough to hold relaying nodes.

This increase rebroadcast probability when the number of vehicles decreases in increasing road segment in the last slot, it clearly notices that the redundancy ratio decrease better in EEAPD compares to other protocols, and vice versa.

Moreover, Figure 10 shows an important result of EEAPD compares to other protocols. It shows clearly that when the road segment increases to 450, the message delivered by EEAPD is around 9.8 (98%) which approximates to 1 (100%) of the packet delivery ratio. When S1PD, SEAD, delay-based and AddP at road segments increase from 360 to 450, the packet delivery ratio starts moves to steady state (Refer Table 3). This show that EEAPD is a suitable choice for the reliability of data dissemination. This helps to minimize the broad storm problem.

Figure 11 presents the comparison among our proposed protocol EEAPD and other approaches such as SEAD, S1PD, delay-based and AddP in how a number of road segments determine the relaying vehicles. In this case, we discuss two points of view, namely the comparison between EEAP and (SEAD and S1PD), and the comparison between EEAPD and (delay-based and AddP).

EEAPD versus (SEAD and S1PD): it is clearly seen from Figure 11 that EEAPD perform better than (SEAD and S1PD) for all road segments. It is interesting that the percentage of relaying vehicles is directly proportional to road segments. The percentage of EEAPD relaying vehicles increases better with an increasing number of road segments. EEAPD receives almost 70% of relaying vehicles to compare to other protocols. When road segments (slots) decrease also the percentage of the relaying node decreases for both protocols, but it is observed that the percentage of EEAPD relaying vehicles devalues better than SEAD and S1PD considering the number of the road segment is 150. This means that the proposed framework has a higher percentage of reducing the number of relaying vehicles when the number of road segments increases compare to SEAD and S1PD protocols. This observation

shows that the road segment will consider the best choice of relaying node for message rebroadcast at the cost of dissemination delay.

EEAPD versus (Delay-Based and AddP): EEAPD performs better than other protocol approaches by 70% when the road segments are 450. This is for two reasons. First, the AddP protocol takes consideration of beacon exchange such as neighbors' speed, angle, position etc. before rebroadcast this can lead delay of data rebroadcasting while EEAPD considers only the surrounding vehicle density as one of the parameters on making rebroadcasting decision no need of beacon exchange. Second, EEAPD has a strategy to maintain the usage of network bandwidth by reducing the relaying vehicles while increasing the road segments in communication range. Therefore, the network resource i.e., bandwidth consumption will significantly decrease. This helps to minimize the broad storm problem.

Figure 12 presents end-to-end delay comparison performance between protocols. As known the transmission delay is a very important metric for the reliability of safety data dissemination in the vehicular environment, EEAPD outperforms than other protocols approaches on time is taken by vehicle to rebroadcast the message and received at the destination in an urban environment. Due to vehicles being close together in the road segment the delay present is constant to all protocols by increasing road segments. Specifically, EEAPD has reduced the transmission delay to around 50 ms compare to S1PD where its transmission is approximate to 70 ms. This observation shows that the road segment will consider the best choice of relaying node for message rebroadcast at the cost of dissemination delay.

Moreover, under various road segments (slots) as depicted in Figures 9–12 it illustrates an important feature of road segments. The more road segments increase the more best relaying node for transmitting a message is selected. EEAPD is better than SEAD, S1PD, delay-based and AddP in term of reducing the number of relaying vehicles involving in the data dissemination process when the number of road segments increases. Therefore, this clarifies the impact of the road segments of EEAPD in a real-time environment. To examine the performance evaluation of the proposed protocol, the simulation is conducted under a high number of road segments (450) in order to analyze the adaptability in a realistic environment feature. The brief summary of various performance metrics according to the impact of road segments in a realistic environment is presented in Table 3.

## 4.4. Impact of Vehicle Density in Reliability of Data Dissemination

In this section, the effects of vehicle density with protocol performance metrics were evaluated when configuring the EEAPD protocol in a real-time network environment that can be adopted for data safety applications.

As depicted from Figure 13, when the number of vehicles becomes higher the forwarding ratios devalue in both protocols. The result shows that EEAPD forwarding ratio degrades better than other protocols on forwarding vehicle selection or reducing unnecessary broadcast within the transmission range (area of interest). As expected, in the EEAPD protocol the number of forwarder vehicles decreased around 65% in dense network compared to AddP, reduced by 45%, SEAD reduced by 50%, delay-based reduced by 40%, and S1PD reduced by 25% which is performed poor compare to all protocols because of receiving a large amount of redundant messages (refer to Figure 2). The better selection of relaying vehicles the more the probability of broadcast performance is better in EEAPD.

As portrayed in Figure 14 both S1PD, SEAD, AddP, and delay-based obtained an approximately equal amount of packet delivery ratio at increasing vehicle density from 50 to 100 while the packet delivery ratio of EEAPD is approximately 1 (100%), (refer to Table 4). This show that EEAPD outperforms all protocols by delivering the messages while optimizing the network bandwidth. One requirement for a better data dissemination protocol is to archive the packet delivery ratio close to 100%. But increasing the forwarder nodes may lead to overload link bandwidth, hence network contention and collision may occur as in Figure 15 which illustrates the amount of packet data carried per time ("link-load" (bits/s)) by EEAPD and those carried by other protocol. Here EEAPD protocol performs better than SEAD, S1PD, AddP and delay-based protocols for all vehicles' density. EEAPD

has lower the link load to 60% compare to other protocols (delay-based = 68%, AddP = 74%, SEAD = 80% and S1PD is more than 100%) at 100 Veh/km. This demonstrates that EEAPD protocol able to adapt particularly in a high-density network with respect to redundant transmission obtained. Figure 2 presents the redundancy ratio doesn't exceed 3 at vehicles' density is 63 vehicles/km when increasing in vehicles' density. This happens via efficient and effective use of bandwidth that enables different applications to be performed simultaneously by increase network capacity.

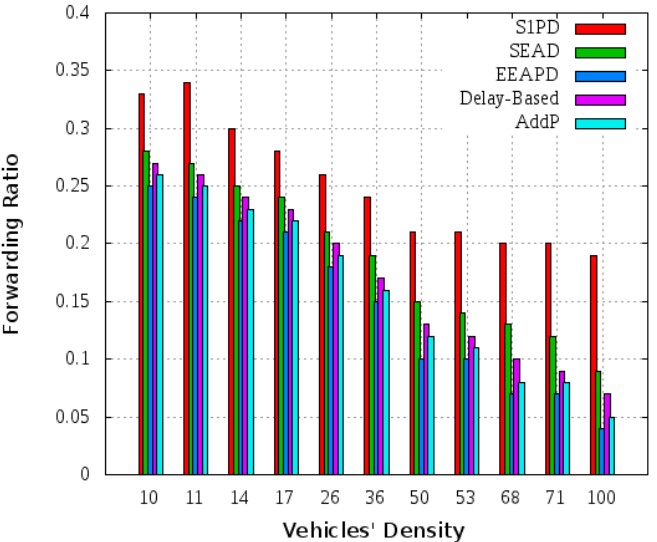

**Figure 13.** Forwarding ratio vs vehicle density.

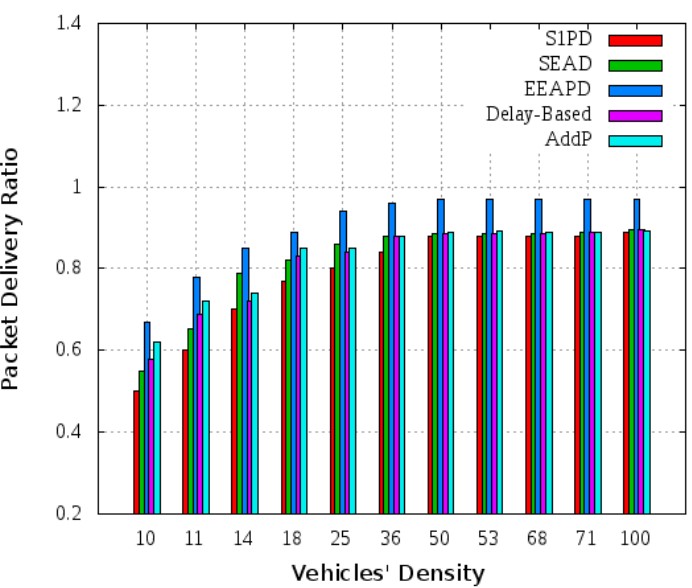

**Figure 14.** Packet Delivery Ratio Vs Vehicles' Density.

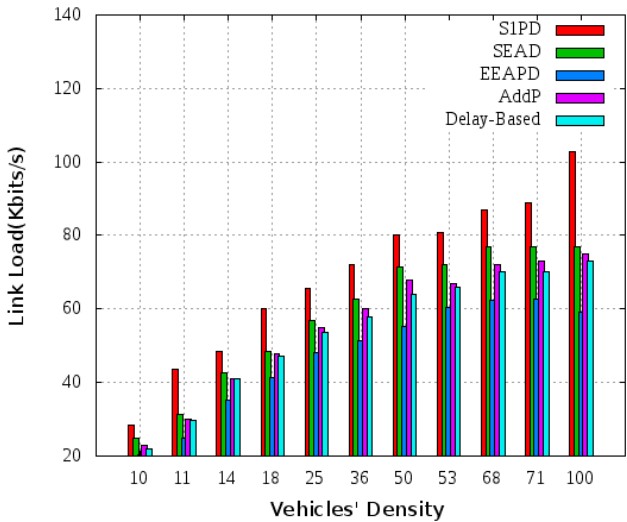

**Figure 15.** Link load vs. vehicle density.

**Table 4.** Protocols comparison based on vehicle density.

| *PROTOCOL* | Rr% | PDr% at 100 Veh/km | Fr% | Ll% | Pdr% at 100 Veh/km | Rv% | E2E (ms) at 10 Veh/km |
|---|---|---|---|---|---|---|---|
| S1PD | 53.8 | 89 | 25 | 102 | 55 | 72 | 40 |
| SEAD | 39.5 | 89.5 | 50 | 80 | 38 | 60 | 50 |
| Delayed-Based | 36 | 89.5 | 40 | 68 | 36 | 60 | 60 |
| AddP | 34 | 89.2 | 45 | 74 | 34 | 55 | 70 |
| **EEAPD** | **30** | **97** | **65** | **60** | **27** | **50** | **80** |

KEY: **Rr**: Redundancy ratio; **PDr**: Packet Delivery ratio; **Rv**: Relaying vehicles; **E2E**: End-to-End delay; **Ll**: Link Load (bits/s); **Pdr**: Packet drop ratio; **Fr**: Forwarding ratio.

Furthermore, as depicted in Figure 16, EEAPD shows a better result by archiving a better packet drop ratio of about 27% compared with other protocols (AddP = 34%, Delay-Based = 36, SEAD = 38 and S1PD = 55%); this observation shows that how EEAPD protocol handles broadcast storm problem in VANET by reducing message collision, message redundancy, erroneous messages and network contention when comparing with SEAD, AddP, S1PD, and belay-based protocols while archiving low end-to-end delay as portrayed in Figure 17.

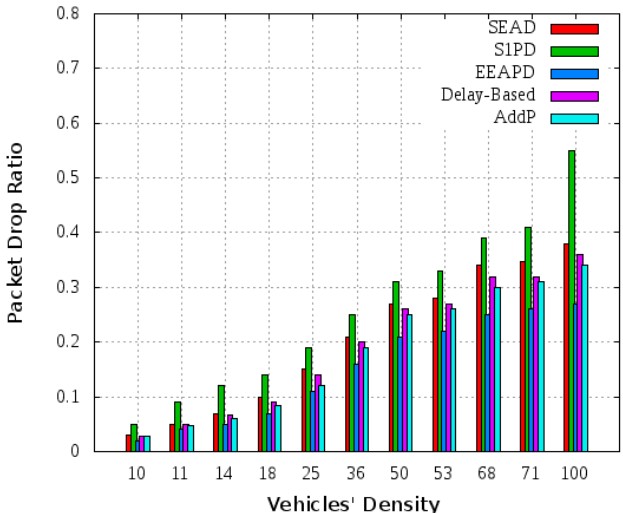

**Figure 16.** Packet Drop Ratio Vs Vehicles' Density.

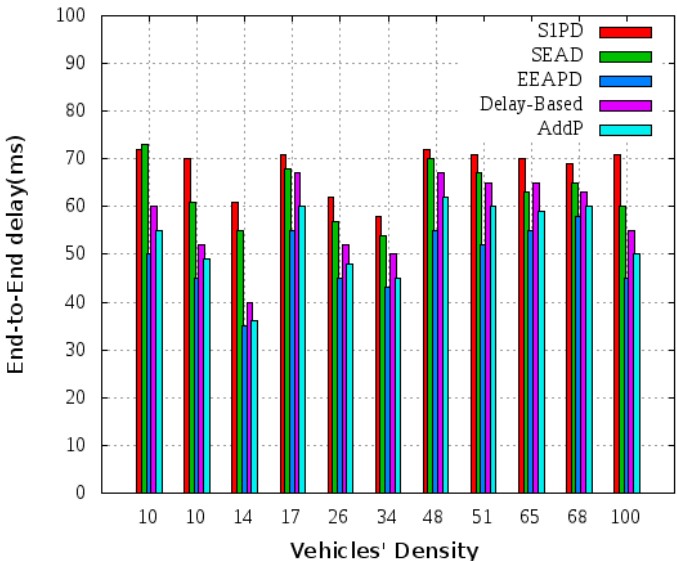

**Figure 17.** End-to-end vs. vehicle density.

Figure 18 show the percentage of relaying vehicles with different vehicles densities. When the vehicle density decreases, the percentage of relaying vehicle increase in a dense network. EEAPD archive 80% better-relaying vehicles at vehicle density are 10 and perform poorly when vehicle density increase to 100 compare to SEAD, S1PD, AddP, and Delay-Based protocols. This shows that not only when the vehicle density is small it helps to reduce the broadcast storm problem but also increases the forwarding ratio (refer to Figure 13). The brief summary of various performance metrics according to the impact of vehicle density in a realistic environment is presented in Table 4.

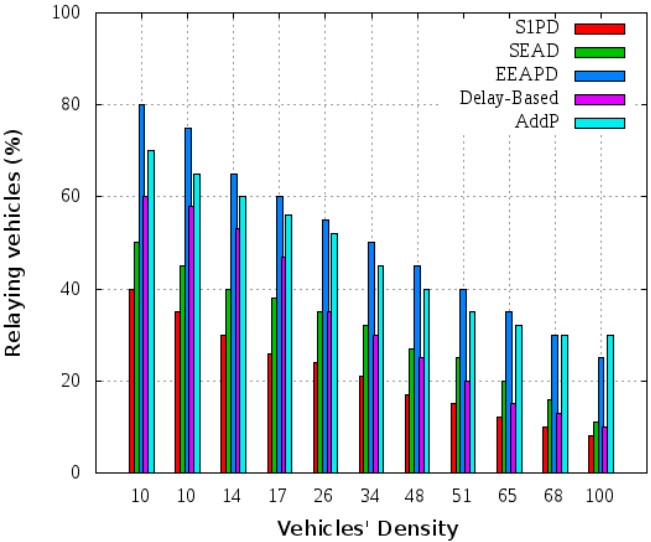

**Figure 18.** Relaying vehicles vs vehicle density.

The benefit for this proposed EEAPD protocol is that each vehicle will be chosen as suitable node among other nodes to be forwarder node according to five important factors: number of road segments within transmission range, distance from accident node, minimum waiting time to rebroadcast, message direction (further behind the receiver vehicle), and the nature of current network density. Therefore, each vehicle will determine the best decision for rebroadcasting the packet without the knowledge from neighboring nodes. Tables 3 and 4 give the performance metrics comparison between

EEAPD protocol (highlighted) and other protocols. This indicates that the performance of the proposed protocol is better than others in terms of all metrics.

## 5. Concluding Remarks and Future Work

In this work, an accurate hybrid based dissemination protocol was designed called EEAPD. The goal is to minimize the so-called "broadcast storm problem". The EEAPD protocol is designed to be more effective and efficient only for safety applications. For this circumstance, EEAPD is considered as beaconless scheme and takes into account the number of road segments in determined which node is suitable for rebroadcasting the message. By using the "redundancy ratio" metric, each node is able to identify the surrounding neighboring vehicles' density. With the proposed protocol, simulation results show the proposed protocol performed better than SEAD, AddP, delay-based and S1PD considered some performance metrics in which EEAPD provides a high packet delivery ratio with low end-to-end delay since it reduces the number of forwarder nodes (reduce broadcast messages) hence "link load" by default reduced and increase network capacity. The findings show that the proposed protocol configuration setting is practically achieved in realistic vehicular environment architecture (V2V) within a fixed number of road segments. In a realistic environment when the road segments increase, more network topology is connected hence the reliability of data dissemination operates simultaneously in the safety application. The simulation was performed to prove the effectiveness and efficiency of the proposed protocol in a realistic environment. Future work includes the EEAPD protocol in a scattered network since there is frequently a vehicle connectivity problem and the use of mathematical models to address the EEAPD parameterization.

**Author Contributions:** Conceptualization, J.S.; formal analysis, S.H. and T.T.; funding acquisition, T.T.; methodology, J.S.; project administration, D.W.; software, J.S.; supervision, D.W.; validation, S.H.; writing—original draft, J.S.; writing—review and editing, J.S.

**Funding:** This work is supported by the National Natural Science Foundation of China (61370201), Open Research Found from the Key Laboratory for Computer Network and Information Integration (Southeast University, Ministry of Education, China).

**Acknowledgments:** I would like to take this opportunity to thank all anonymous reviewers and editor for their useful comments/suggestions and contributions of this paper. I also thankful to my friends who help me a lot in the completion of this paper.

**Conflicts of Interest:** The authors declare no conflict of interest.

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
