# Peer review of "An Effective and Efficient Adaptive Probability Data Dissemination Protocol in VANET"

_data, 2017_

Round 1
Reviewer 1 Report
The authors study the VANET networks and they propose an effective and efficiency adaptive probability data dissemination protocol named “EEAPD” which is comprised of delay scheme and probabilistic approach. The topic of paper is interesting and the authors have well-thought out their main contributions. The proposed framework can improve the existing data dissemination in VANETs and support their smooth operation. Some minor comments to improve the quality of presentation of the paper are as follows.
Given that the VANETs are adhoc networks the authors should better review the existing literature regarding the approaches that have been proposed to support the data dissemination in adhoc networks which mainly propose clustering mechanisms based on nodes social characteristics (e.g., "Interest, energy and physical-aware coalition formation and resource allocation in smart IoT applications." In Information Sciences and Systems (CISS), 2017 51st Annual Conference on, pp. 1-6. IEEE, 2017, "Interest-aware energy collection & resource management in machine to machine communications." Ad Hoc Networks 68 (2018): 48-57) and physical characteristics like mobility, position, distance, etc. (e.g., "Adaptive clustering for mobile wireless networks." IEEE Journal on Selected areas in Communications 15, no. 7 (1997): 1265-1275, “Routing in clustered multihop, mobile wireless networks with fading channel." In proceedings of IEEE SICON, vol. 97, no. 1997, pp. 197-211. 1997).
The authors should provide a justification and discussion in section 2, update the provided literature review and list of references and better correlate their work to the adhoc nature of the VANETs.
Another main concern that is raised by the manuscript and the provided analysis, is the complexity and implementation cost of the proposed approach. The authors should provide some additional comparative numerical results to other relevant research works from the recent literature in order to show the benefit of adopting their proposed framework. The presented numerical results show mainly the pure performance of the proposed framework, however without a comparative study to other approaches, the reader is not able to understand the benefits of the proposed framework. The authors should justify how the proposed framework can be implemented in a real-time manner within the existing networking environment. Furthermore, a minor comment: the authors should carefully check the manuscript, as there are several typos and improve the quality of figures.
Author Response
Dated 15-12-2018
Dear Reviewer.
On behalf of the co-authors, I would like to acknowledge for giving us the comments of our manuscript data-405527. Please see our point-by-point response to the comments below. We have modified the manuscript data-405527 according to suggestions and advice which you gave us and believe the manuscript data-405527 has significantly improved. We hope it will be suitable for publication.
Yours sincerely,
Sospeter John
Point 1: Given that the VANETs are ad-hoc networks the authors should better review the existing literature regarding the approaches that have been proposed to support the data dissemination in ad-hoc networks which mainly propose clustering mechanisms based on nodes social characteristics (e.g., "Interest, energy, and physical-aware coalition formation and resource allocation in smart IoT applications." In Information Sciences and Systems (CISS), 2017 51st Annual Conference on, pp. 1-6. IEEE, 2017, "Interest-aware energy collection & resource management in the machine to machine communications." Ad Hoc Networks 68 (2018): 48-57) and physical characteristics like mobility, position, distance, etc.(e.g., "Adaptive clustering for mobile wireless networks." IEEE Journal on Selected Areas in Communications 15, no. 7 (1997): 1265-1275, “Routing in clustered multihop, mobile wireless networks with fading channel." In Proceedings of IEEE SICON, vol. 97, no. 1997, pp. 197-211. 1997).
Response 1: we have gone through the provided literature about data dissemination in ad hoc networks regarding clustering mechanisms based on;
i. Nodes social characteristics:
(a) "Interest, energy and physical-aware coalition formation and resource allocation in smart IoT applications." In Information Sciences and Systems (CISS), 2017 51st Annual Conference on, pp. 1-6. IEEE, 2017.
(b) "Interest-aware energy collection & resource management in the machine to machine communications." Ad Hoc Networks 68 (2018): 48-57)
The discussion about social characteristics found on above literature has been explained in section 2 [manuscript data-405527, page 6, line 215-222] also they have been included in the reference list.
ii. Nodes physical characteristics:
(c) "Adaptive clustering for mobile wireless networks." IEEE Journal on Selected Areas in Communications 15, no. 7 (1997): 1265-1275,
(d) “Routing in clustered multihop, mobile wireless networks with fading channel." In Proceedings of IEEE SICON, vol. 97, no. 1997, pp. 197-211. 1997.
The discussion about physical characteristics found on above mention literature has been explained in section 2 [manuscript data-405527, page 4-5, line 144-157] and have been included in the reference.
Point 2: The authors should provide a justification and discussion in section 2, update the provided literature review and list of references and better correlate their work to the ad-hoc nature of the VANETs.
Response 2: According to manuscript data-405527 in section 2, we have provided the justification and discussion about the relevant works that are correlated to the VANET approach. We have found two additional recent literature works that match with our framework about data dissemination in VANET.
Point 3: Another main concern that is raised by the manuscript and the provided analysis, is the complexity and implementation cost of the proposed approach. The authors should provide some additional comparative numerical results to other relevant research works from the recent literature in order to show the benefit of adopting their proposed framework. The presented numerical results show mainly the pure performance of the proposed framework, however, without a comparative study to other approaches, the reader is not able to understand the benefits of the proposed framework. The authors should justify how the proposed framework can be implemented in a real-time manner within the existing networking environment. Furthermore, a minor comment: the authors should carefully check the manuscript, as there are several typos and improve the quality of figures.
Response 3: In this part, we did the following;
i. Additional comparative numerical results.
· We study some of the relevant research works from recent literature and discussed them in manuscript data-405527, section 2.
· We provided the comparison of related works according to different characteristic namely; (a) application type (b) forwarder parameter (c) simulated scenario (d) beacon-based and (e) Communication mode. This can be found in [manuscript data-405527, Table 1., page 7, line 245]
· Then we provide additional comparative numerical results apart from the previous experiment conducted in order to compare and show how EEAPD protocol is benefit from other provided works. The discussion about comparative numerical results can be found in [manuscript data-405527, section 4.]
· The brief summary of comparison between the proposed protocol [EEAPD] and other protocols [S1PD, SEAD, Daley-Based and AddP] according to the number of road segments and vehicles’ density also found in [manuscript data-405527, Table 3. and Table 4. respectively]
ii. How the proposed framework can be implemented in a real-time manner within the existing networking environment.
· We thank for the design of the probability formula for broadcasting data [Refer equation (3)], each node can able to decide when to rebroadcast the accident message depend on the current state of the network. The performance of the EEAPD protocol to the networking realistic environment is adapted to the safety application requirement by considering two parameters namely “road segments” and “vehicle density”. For better proof the correctness of EEAP adaptability in a realistic networking environment, we demonstrate the proposed protocols under both parameters. In which the higher data reachability obtained around 98% and 97% on both road segments and vehicles’ density respectively while saving network capacity and limited bandwidth consumption
iii. Typos and improve the quality of figures.
· We have proofread and eliminated some typos found throughout of manuscript-data-405527 as well as we have improved the quality of figures and add more figures according to the experiment conducted and results obtained again in order to provide the good performance comparison between the proposed framework and other works to a reader as it can be seen in manuscript-data-405527

Reviewer 2 Report
The paper deals with a problem that is being studied for quite a while, but there is a contribution here. Despite presenting a relevant work, some issues must be addressed before the publication.
1) The title is not good! The word “Efficiency” is not correct here… you mean “efficient”
2) The authors should proofread to eliminate typos that can be found here and there along the text. Moreover, please, refrain from using the first person, “we, our, us”, please prefer using third person or passive voice instead.
3) I found a relevant related work published in the journal Ad Hoc Networks in 2013 that must be included in your references. The title is:
“Exploring geographic context awareness for data dissemination on mobile ad hoc networks”.
Notice that the authors provide simulations with different types of maps, and the traces used by them are provided by an open repository. Please, justify why you didn’t use traces from others and collect your own.
4) Section 1 does not need subsections. The text can flow without these divisions.
5) In the introduction, it is necessary that the authors clearly state the contributions of their work.
6) Improve resolution of figures 4, 5 and 7. The current quality is bad.
7) The word “safety” is used with a wrong meaning in Section 4.2. What you mean there is “reliability”. There are other works along the text that are “weirdly” used. I strongly suggest a professional revision of the text regarding the usage of the language.
Author Response
Point 1: The title is not good! The word “Efficiency” is not correct here… you mean “efficient”
Response 1: We acknowledge the mistake concerning our “TITLE”. We have corrected this title in the manuscript in Page 1, Line 2 “Effective and Efficient Adaptive Probability Data Dissemination Protocol in VANET. Apart from the heading-title, also we have corrected the word “efficiency” to “efficient” where the title is mentioned [manuscripts data-405527, Page 1 Line 15 and Page 7 Line 267]
Point 2: The authors should proofread to eliminate typos that can be found here and there along the text. Moreover, please, refrain from using the first person, “we, our, us”, please prefer using third person or passive voice instead.
Response 2:
i. We have proofread and eliminated some typos found throughout of manuscript-data-405527
ii. We have voided from using the first person, instead, we have used passive voice and third person throughout of manuscript-data-405527
Point 3: I found a relevant related work published in the journal Ad Hoc Networks in 2013 that must be included in your references. The title is:
“Exploring geographic context awareness for data dissemination on mobile ad hoc networks”.
Notice that the authors provide simulations with different types of maps, and the traces used by them are provided by an open repository. Please, justify why you didn’t use traces from others and collect your own.
Response 3:
i. We found this work namely; “Exploring geographic context awareness for data dissemination on mobile ad hoc networks” is relevant related to our work, so we have discussed in [manuscripts data-405527, section two, page 5, line 158-173 references [48].
ii. We conducted simulation using a real-world map [Refer Figure 8.] in the manuscripts data-405527 page 14, line 443-452 which is a part of the Dalian city (Dalian University of Technology (DUT) area). This map is extracted from OpenStreetMap Online at https://www.openstreetmap.org/ whereby this is an open-access website that keeps all data found in a specific area such as road, railway, roadside infrastructure and trails from all over the world. Figure 8(a) and Figure 8(b) are the same extracted map whereby Figure 8(a) show really different types of data found while Figure 8(b) just present the standard map, however during simulation we consider them as a single map to generate realistic mobility via SUMO-Simulator.
iii. We collect our own map (Dalian University of Technology map (DUT)) because of the following reasons:
· DUT is one of the busiest areas in Dalian City where we simulated huge number vehicle density, road lanes, different types of traffic data and road pattern so that results from simulation correctly reflect the real-world performance of a VANET e.g. vehicles allocated to streets are separated by building, trees and other objects.
· By using our own map, we investigated the effect of vehicles density with sufficient level of a number of road segments (slots) it gives an accurate network simulation result compare to trace map from others.
· Due to ITS suffer from network disconnection, we prefer to use our own map where vehicles are accumulated in a good manner so that we cannot experience too many networks disconnected in a simulated and realistic environment.
Point 4: Section 1 does not need subsections. The text can flow without these divisions.
Response 4: we have arranged section 1 namely “Introduction” without any subsection in the manuscript, we have removed subsection heading namely “Communication”, “Characteristics”, “Application” and “Data dissemination”. Now the introduction section flows smoothly without subsection in [manuscripts data-405527, page 1, line 29-105].
Point 5: In the introduction, it is necessary that the authors clearly state the contributions of their work.
Response 5: We have clearly stated the contributions of our work in the manuscript page 3, line 81-102 The main contributions of this work are fourfold as follows:
EEAPD delivers an effective and efficient data dissemination protocol. The proposed protocol is more useful because it considers the critical parameters such as the number of road segments (slots), minimal waiting time, number of vehicles (vehicles’ density), and message direction in rebroadcasting probability decision. EEAPD ensures effectiveness and efficiency by not only evaluating the low End-to-End delay buts also high packet delivery. Additionally, it offers an effective delivery of a message for the nodes’ receiver to understand, interpret and make use of it as intended. Nevertheless, EEAPD protocol reduces bandwidth consumption.
EEAPD provides the ability to adapt to the protocol behavior efficiently for safety applications’ requirements. This protocol is an adaptive probability protocol that offers efficiency and effectivenesses for safety application. This application is intended to ensure the passengers’ protection from danger, risk or injury on roads.
EEAPD offers the ability to adopt an environment with no beacon exchange. The protocol depends only on surrounding vehicles’ density.
The simulation of EEAPD considered the number of road segments and vehicles’ density. EEAPD was compared with beaconless protocol and beacon-based protocol. Results show that EEAPD enhanced the performance of data dissemination in a realistic environment
Point 6: Improve resolution of figures 4, 5 and 7. The current quality is bad.
Response 6: We acknowledge the suggestion you gave us to improve the resolution of figure 4, 5 and 7. We have improved the figures’ mentioned by increasing the resolution, now in manuscripts data-405527 the figures look good as can be seen;
Figure 4. Receiving procedure of EEAPD. [page10, line 354-357]
Figure 5. Rebroadcasting procedure of EEAPD [page 11, line 364-367]
Figure 7. The relationship between Vehicles’ density, Redundancy ratio and the probability of rebroadcast. [page 13, line 406-409]
Point 7: The word “safety” is used with the wrong meaning in Section 4.2. What you mean there is “reliability”. There are other works along the text that are “weirdly” used. I strongly suggest a professional revision of the text regarding the usage of the language.
Response 7: We acknowledge the advice regarding the use of the language in this section.
i. We changed the word “safety” to “reliability” in section 4.2.1 regarding its heading-title. From “Impact of Road Segments in Safety Data Dissemination” to “Impact of Road Segments in the Reliability Data Dissemination” in the [manuscripts data-405527, page 16, line 501].
ii. We did the revision regarding the usage of language throughout in the manuscript-data-405527

Round 2
Reviewer 2 Report
I think the authors sufficiently addressed my comments and suggestions. The paper is good now for publication.